# A Supervised Speech Enhancement Approach with Residual Noise Control for Voice Communication

**Andong Li [1,2], Renhua Peng [1,2] and Chengshi Zheng [1,2,*] and Xiaodong Li [1,2]**

[1]  Key Laboratory of Noise and Vibration Research, Institute of Acoustics, Chinese Academy of Sciences, Beijing 100190, China; liandong@mail.ioa.ac.cn (A.L.); pengrenhua@mail.ioa.ac.cn (R.P.); lxd@mail.ioa.ac.cn (X.L.)

[2]  University of Chinese Academy of Sciences, Beijing 100049, China

*  Correspondence: cszheng@mail.ioa.ac.cn

**Abstract:** For voice communication, it is important to extract the speech from its noisy version without introducing unnaturally artificial noise. By studying the subband mean-squared error (MSE) of the speech for unsupervised speech enhancement approaches and revealing its relationship with the existing loss function for supervised approaches, this paper derives a generalized loss function that takes residual noise control into account with a supervised approach. Our generalized loss function contains the well-known MSE loss function and many other often-used loss functions as special cases. Compared with traditional loss functions, our generalized loss function is more flexible to make a good trade-off between speech distortion and noise reduction. This is because a group of well-studied noise shaping schemes can be introduced to control residual noise for practical applications. Objective and subjective test results verify the importance of residual noise control for the supervised speech enhancement approach.

**Keywords:** generalized loss function; residual noise control; noise shaping; speech distortion; deep learning

---

## 1. Introduction

Speech enhancement plays an important role in noisy environments for many applications, such as speech communication, speech interaction and speech translation. Numerous researchers have spent much effort on separating the speech from its noisy version and various approaches have already been proposed in the last five decades. Conventional approaches include spectral subtraction [1], statistical method [2,3] and subspace-based method [4], which has proved to be valid when the additive noise is stationary or quasi-stationary. However, their performance often suffers from heavy degradation under non-stationary and low signal-to-noise ratio (SNR) conditions.

Recently, supervised deep learning approaches have shown their powerful capability on suppressing both stationary and highly non-stationary noise signals, which is mainly because of the highly nonlinear mapping ability of deep neural networks (DNNs) [5–9]. In DNN-based algorithms, minimum mean-squared error (MMSE) is often adopted as a loss criterion to update the weights of the network. Nevertheless, the usage of this criterion directly may suffer from some problems. First, although MSE is the most popular and well-known criterion, it is based on the assumption, i.e., each time-frequency (T-F) bin bears the same importance during the training phase. However, the auditory system has different sensitivity towards different frequency regions and the perception knowledge should be also taken into account [10,11]. Second, the speech spectrogram has unbalanced distribution, i.e., the formants often exist in the low- and middle-frequency regions while they are sparse in the high-frequency regions. Therefore, global MSE optimization usually obtains an

over-smoothing estimation which omits some important detailed information. To solve these problems, many new criteria, that consider speech perception, have been proposed in recent years [12–15]. The first one is to use perceptually weighted MSE functions, which are proposed to weight the loss in different T-F regions [13,16]. Despite the weighted coefficients mitigate the over-smoothing issue of MSE to some extent, most of them are based on the heuristic principles and cannot solve the inherent over-smoothing problem of MSE completely. The second one is to use objective metrics as loss functions. For examples, perceptual evaluation speech quality (PESQ) [17], short-time objective intelligibility (STOI) [18] and scale-invariant speech distortion ratio (SI-SDR) [19] have been adopted as loss functions. Although the metric-based methods facilitate the better optimization of the specific objective metric, other metrics are often non-optimized. Moreover, the calculation of some metrics is often too complicated and non-continuous [20]. In [21], speech distortion and residual noise are considered separately in the loss function, known as the components loss (CL), which obtains relatively better metric scores than MSE when suitable loss-weighted coefficients are selected.

Note that all the above mentioned loss functions aim at suppressing noise as much as possible at noise-only segments. In other words, at noise-only segments, the amount of noise reduction is expected to be a positive infinite value. However, this aim could not be achieved in most cases for many reasons. First, the noise is often stochastic, and thus it is inevitable that the estimation accuracy is often constrained by a limited number of available observations [22,23]. Second, there are a great variety of noise signals, so that a DNN model cannot be expected to distinguish all of them correctly from the speech in each T-F unit. Therefore, when the noise cannot be suppressed totally as expected, some unnatural residual noise may severely degrade speech quality [24], which needs to be considered carefully. In this paper, we derive a generalized loss function by introducing multiple manual parameters to flexibly make a balance between speech distortion and noise attenuation. More specifically, we use the residual noise control introduced for voice communication [25,26]. By theoretical derivations, MSE and some other often-used loss functions can be included in the proposed generalized loss function.

The remainder of the paper is structured as follows. Section 2 formulates the problem. Section 3 derives the generalized loss in detail and introduces used network architecture. Section 4 is the experimental settings. Results and analysis are given in Section 5. Section 6 presents some conclusions.

## 2. Problem Formulation

In the time domain, the noisy signal can be modelled as

$$x\left(n\right) = s\left(n\right) + d\left(n\right), \tag{1}$$

where $s\left(n\right)$ is the clean speech and $d(n)$ is the additive noise. In the frequency domain, (1) can be written as

$$X_l\left(k\right) = S_l\left(k\right) + D_l\left(k\right), \tag{2}$$

where $X_l\left(k\right)$, $S_l\left(k\right)$, and $D_l\left(k\right)$ are, respectively, discrete Fourier transforms (DFT) of $x(n)$, $s(n)$, and $d(n)$ with the frame index $l$ and the frequency bin $k$.

For practical applications, we only have the time-domain noisy signal $x(n)$ or its frequency-domain version $X_l(k)$, the problem becomes how to estimate $s(n)$ or $S_l(k)$ from its noisy signal. It is common to use MMSE as a criterion in unsupervised speech enhancement approaches. Before introducing MMSE, we first define the square error as

$$J_x\left[M_l\left(k\right)\right] = \left|f\left(S_l\left(k\right)\right) - g\left(S_l\left(k\right), D_l\left(k\right), M_l\left(k\right)\right)\right|^2, \tag{3}$$

where $M_l\left(k\right)$ is a nonlinear spectral gain function, $f\left(a\right)$ and $g(a,b,c)$ are the transform functions. When $f\left(a\right) = |a|$ and $g(a,b,c) = |(a+b)c|$, $\min_{M_l(k)} E\left\{J_x\left[M_l\left(k\right)\right]\right\}$ results in MMSE spectral amplitude estimator in [2], where $E\left\{\bullet\right\}$ is the expectation operator. When $f\left(a\right) = \log(|a|)$ and $g(a,b,c) = \log(|(a+b)c|)$,

$\min\limits_{M_l(k)} E\{J_x[M_l(k)]\}$ leads to MMSE log-spectral amplitude estimator in [3]. More complicated forms of $f(a)$ and $g(a,b,c)$ can be chosen; for example, many perceptually-weighted error criteria can be included, which can be referred to [10].

For supervised approaches, the square error in the subband is often defined as the loss function in the fullband, which is

$$\mathcal{J}_x = \sum_{k\in\mathcal{K}}\sum_{l\in\mathcal{L}} J_x[M_l(k)]. \tag{4}$$

One can get that, when $f(a) = \log(|a|)$ and $g(a,b,c) = \log(|(a+b)c|)$, $\min\limits_{M_l(k)}\{\mathcal{J}_x\}$ is to minimize the MSE of log-spectral amplitude between the clean speech and the estimated speech, which is the training target in [6].

Note that (3) and (4) are quite similar and the most obvious difference between them is that $J_x[M_l(k)]$ is the subband square error, while $\mathcal{J}_x$ is the fullband square error. The other difference is that the nonlinear spectral gain can be derived theoretically by minimizing $E\{J_x[M_l(k)]\}$ when the probability density function (p.d.f.) of the speech and that of the noise are both given, while it is difficult to derive the nonlinear spectral gain by minimizing $\mathcal{J}_x$, where this gain can often be mapped from the input noisy features after training the supervised machine learning model. In all, it seems that all subband square error functions can be generalized to the fullband ones as supervised training targets.

In the above formulation, MMSE is utilized to optimize the speech spectrum recovery in the T-F domain. However, due to the stochastic characteristic of noise components and the performance limit of the network, the residual noise tends to be unnatural and may severely degrade the speech quality. Moreover, noise suppression and speech distortion are not separately considered in an explicit way, which motivates us to reformulate the optimization towards the trade-off between noise suppression and speech distortion and derive a new type of generalized loss function in Section 3.

## 3. Proposed Algorithm

Only using MMSE as a criterion, it is difficult to make a balance between speech distortion and noise reduction. This section derives a more generalized fullband loss function.

### 3.1. Trade-Off Criterion in Subband

In traditional speech enhancement approaches, speech distortion and noise reduction in the subband can be considered separately. The subband square error of the speech and the subband residual noise can be, respectively, given by

$$J_s[M_l(k)] = |f(S_l(k)) - g(S_l(k), D_l(k), M_l(k))|^2, \tag{5}$$

and

$$J_d[M_l(k)] = |h(S_l(k), D_l(k), M_l(k))|^2, \tag{6}$$

where $h(a,b,c)$ is a transform function. When $f(a) = |a|$, $g(a,b,c) = |ac|$, and $h(a,b,c) = |bc|$, $E\{J_s[M_l(k)]\}$ and $E\{J_d[M_l(k)]\}$ become the MSE of the speech magnitude and the residual noise power in the subband, respectively, which are identical with [27] (8.31) and (8.32).

By minimizing the subband MSE of the speech with a residual noise control, an optimization problem can be given to derive the nonlinear spectral gain, which is given by

$$\min\limits_{M_l(k)} E\{J_s[M_l(k)]\},$$
$$s.t. \quad E\{J_d[M_l(k)]\} = |\lambda(\beta_l(k), D_l(k))|^2, \tag{7}$$

where $\lambda(\beta, b)$ is a transform function. $\beta_l(k) \in [0\ 1]$ could be both a frequency and frame-dependent factor that can be introduced to control the residual noise flexibly.

The optimal spectral gain in (7) can be solved theoretically by the Lagrange multiplier method, which is

$$\min_{M_l(k)} \left\{ \begin{array}{c} E\left\{J_s\left[M_l\left(k\right)\right]\right\} + \mu E\left\{J_d\left[M_l\left(k\right)\right]\right\} \\ -\mu \left|\hbar{\lambda}\left(\beta_l\left(k\right), D_l\left(k\right)\right)\right|^2 \end{array} \right\}, \tag{8}$$

where $\mu \geq 0$ is a Lagrange multiplier. When $f(a) = |a|$, $g(a,b,c) = |ac|$, $h(a,b,c) = |bc|$, and $|\hbar{\lambda}(\beta, b)| = \beta E\{|b|^2\}$, the optimal spectral gain can be derived from (8) and the constraint in (7), which can be given by

$$M_l(k) = \xi_l(k) / (\xi_l(k) + \mu_l(k)), \tag{9}$$

where $\xi_l(k) = E\{|S_l(k)|^2\}/E\{|D_l(k)|^2\}$ is the *a priori* SNR. It is not always possible to derive $M_l(k)$ mathematically, especially when $f(a)$, $g(a,b,c)$, $h(a,b,c)$, and $\hbar{\lambda}(\beta, b)$ have very complicated expressions. Moreover, it is difficult to accurately estimate the noise power spectral density in non-stationary noise environments [28–30]. However, it seems that this optimization can be easily solved by supervised approaches. To transfer this problem, we need to define the fullband square error of the speech and the fullband residual noise power to derive the loss function for supervised approaches.

### 3.2. Trade-Off Criterion in Fullband

The fullband MSE of the speech and the fullband residual noise can be, respectively, given by

$$\mathcal{J}_s = \sum_{k=\mathcal{K}} \sum_{l=\mathcal{L}} J_s\left[M_l(k)\right], \tag{10}$$

and

$$\mathcal{J}_d = \sum_{k=\mathcal{K}} \sum_{l=\mathcal{L}} J_d\left[M_l(k)\right]. \tag{11}$$

The loss function without any constraints can be given by

$$\mathcal{J}_x = \mathcal{J}_s + \mu \mathcal{J}_d, \tag{12}$$

where (12) is the same as the newly proposed components loss function as given in [21].

The loss function with residual noise control is

$$\mathcal{J}_x = \mathcal{J}_s + \mu \mathcal{J}_d^{\mathrm{con}}, \tag{13}$$

where

$$\mathcal{J}_d^{\mathrm{con}} = \sum_{k=\mathcal{K}} \sum_{l=\mathcal{L}} \left| J_d\left[M_l(k)\right] - \left|\hbar{\lambda}\left(\beta_l(k), D_l(k)\right)\right|^2 \right|.$$

It is obvious that (13) is a generalization of (12), where (13) reduces to (12) when $\left|\hbar{\lambda}\left(\beta_l(k), D_l(k)\right)\right|^2 \equiv 0$. One can observe that $\beta_l(k)$ is both frequency and frame-dependent, so it can control the residual noise in each time-frequency bin.

### 3.3. A Generalized Loss Function

We further generalize the subband square error in (5) and (6), the square is substituted by a variable $\gamma \geq 0$ and an additional variable $\alpha$ is also introduced on the spectra, then (5) and (6) can be, respectively, given by

$$J_s^{\gamma,\alpha}\left[M_l(k)\right] = \left| f\left(S_l^\alpha(k)\right) - g\left(S_l^\alpha(k), X_l^\alpha(k), M_l^\alpha(k)\right) \right|^\gamma, \tag{14}$$

and

$$J_d^{\gamma,\alpha}\left[M_l(k)\right] = \left| h\left(S_l^\alpha(k), D_l^\alpha(k), M_l^\alpha(k)\right) \right|^\gamma. \tag{15}$$

Analogously, with the residual noise control, the optimization problem in the subband becomes

$$\min_{M_l(k)} E\left\{J_s^{\gamma,\alpha}\left[M_l(k)\right]\right\},$$
$$\text{s.t.} \quad E\left\{J_d^{\gamma,\alpha}\left[M_l(k)\right]\right\} = \left|\hbar\left(\beta_l^{\alpha}(k), D_l^{\alpha}(k)\right)\right|^{\gamma}. \tag{16}$$

By setting $f(a) = |a|$, $g(a,b,c) = |ac|$, $h(a,b,c) = |bc|$, and $\hbar(\beta,b) = (\beta|b|)$, one can derive a generalized gain function with the Lagrange multiplier method, which is

$$M_l(k) = \left(\frac{(\xi_l(k))^{c_1}}{(\mu_l(k))^{(2c_1c_2-1)} + (\xi_l(k))^{c_1}}\right)^{c_2}, \tag{17}$$

where $c_1 = \alpha\gamma/(2\gamma - 2)$ and $c_2 = 1/\alpha$, where (17) is identical to [31] (6). Note that [31] (6) is given intuitively without theoretical derivation. When $\gamma = 2$ and $\alpha = 1$, (17) reduces to (9). When $\gamma = 2$, one can get $M_l(k) = \left((\xi_l(k))^{\alpha}/(\mu_l(k) + (\xi_l(k))^{\alpha})\right)^{1/\alpha}$, which has already been derived and presented in ([31] (22)).

Similarly, the generalized loss function for supervised approaches can be given by

$$\mathcal{J}_x^{\gamma,\alpha} = \mathcal{J}_s^{\gamma,\alpha} + \mu\mathcal{J}_d^{\gamma,\alpha,\text{con}}, \tag{18}$$

where the first item $\mathcal{J}_s^{\gamma,\alpha} = \sum_{k=\mathcal{K}}\sum_{l=\mathcal{L}} J_s^{\gamma,\alpha}\left[M_l(k)\right]$ relates to the fullband speech distortion and the second item $\mathcal{J}_d^{\gamma,\alpha,\text{con}} = \sum_{k=\mathcal{K}}\sum_{l=\mathcal{L}}\left|J_d^{\gamma,\alpha}\left[M_l(k)\right] - \left|\hbar\left(\beta_l^{\alpha}(k), D_l^{\alpha}(k)\right)\right|^{\gamma}\right|$ is introduced to control the residual noise.

Equation (18) is a generalized loss function that includes (12) and (13). This is because (18) reduces to (13) when $\gamma = 2, \alpha = 1$ and it can further reduce to (12) by setting $\left|\hbar\left(\beta_l^{\alpha}(k), D_l^{\alpha}(k)\right)\right|^{\gamma} \equiv 0$. It is interesting to see that (3) also can be separated into two components, where one is the MSE of the speech and the other is related to the residual noise. When $f(a) = a$ and $g(a,b,c) = (a+b)c$, we have

$$E\left\{J_x\left(M_l(k)\right)\right\} = E\left\{J_s\left(M_l(k)\right)\right\} + E\left\{J_d\left(M_l(k)\right)\right\}, \tag{19}$$

where $E\left\{J_s\left(M_l(k)\right)\right\} = |1 - M_l(k)|^2 E\left\{|S_l(k)|^2\right\}$ relates to the power of speech distortion and $E\left\{J_d\left(M_l(k)\right)\right\} = |M_l(k)|^2 E\left\{|D_l(k)|^2\right\}$ relates to the power of residual noise. $E\left\{J_x\left[M_l(k)\right]\right\}$ is a combination of speech distortion and residual noise, so the fullband MSE loss function of a complex spectrum is also a special case of the generalized loss function in (18). If $f(a) = |a|$ and $g(a,b,c) = |(a+b)c|$ are chosen, the decomposition of $E\left\{J_x\left(M_l(k)\right)\right\}$ is more complicated than (19), which will not be further discussed for limited space.

In this paper, we emphasize the importance of introducing the residual noise control. $f(a) = |a|$, $g(a,b,c) = |ac|$, $h(a,b,c) = |bc|$, and $\hbar(\beta,b) = (\beta|b|)$ are applied, although more complicated expressions can be chosen when taking the perceptual quality into account. Accordingly, we have

$$\mathcal{J}_s^{\gamma,\alpha} = \sum_{l=\mathcal{L}}\sum_{k=\mathcal{K}}\left|(1 - M_l^{\alpha}(k))S_l^{\alpha}(k)\right|^{\gamma}, \tag{20}$$

and

$$\mathcal{J}_d^{\gamma,\alpha,con} = \sum_{l=\mathcal{L}}\sum_{k=\mathcal{K}}\left||M_l(k)D_l(k)|^{\alpha\gamma} - |\beta_l(k)D_l(k)|^{\alpha\gamma}\right|, \tag{21}$$

where both $\alpha$ and $\beta_l(k)$ are constant values over frequency for simplicity, that is to say, $\beta_l(k) \equiv \beta_0$ and $\alpha = \alpha_0$ are used in the following. $\alpha_0$ is set to 1 in the major experiments and we will also separately analyze the role of $\alpha$. We study the impact of $\beta_0$, $\mu$, $\gamma$ and $\alpha_0$ on supervised approaches.

## 4. Experimental Setup

### 4.1. Dataset

Experiments are conducted with TIMIT corpus, where 1000 and 200 clean utterances are randomly chosen as the training and the validation datasets, respectively. In total, 125 types of environment noises [6,32] are used for generating noisy utterances under different SNR levels ranging from −5 dB to 15 dB with the interval 5 dB. During each mixing process, a clean utterance is mixed with two types of noises and 5 SNR levels. As a consequence, 10,000 ($1000 \times 5 \times 2$), 2000 ($200 \times 5 \times 2$) noisy-clean pairs are established for training and validation, respectively. For model test, additional 10 male and 10 female utterances are chosen to mix with 5 types of unseen noises taken from the NOISEX92 [33] (babble, factory1, hfchannel, pink, and white) with SNR ranging from −5 dB to 10 dB with the interval 5 dB.

### 4.2. Experimental Settings

We sample all the utterances at 16 kHz, which are subsequently enframed by a 20-ms Hamming window and 10-ms overlap between adjacent frames. A 320-point short-time Fourier transform (STFT) is applied to transform the frames into the T-F domain, leading to 161-point spectral feature vectors. The magnitude of the spectrum is deployed as the input feature. The models are trained with stochastic gradient descent (SGD) optimized by Adam [34]. The learning rate is initialized at 0.0005. We halve the learning rate only when three consecutive validation loss increment arises and the training process is early-stopped unless ten consecutive validation loss increment happens. Totally 100 epochs are trained to guarantee the network convergence. Within each epoch, the minibatch is set to 16 at the utterance level, where all the utterances are zero-padded to have the same timestep with the longest utterance.

### 4.3. Network Architecture

U-Net is chosen as the network in this paper, which has been widely adopted for the speech separation task [35]. As shown in Figure 1, the network consists of the convolutional encoder and decoder, both of which are comprised of five convolutional blocks where the 2-D convolution layer is adopted, followed by batch normalization (BN) [36] and exponential linear unit (ELU) [37]. Within each convolutional block, the kernel size is set to $(2, 3)$ along the temporal and frequency axis. Skip connections are introduced to compensate for the information loss during the features compression process. Note that the mapping target is the gain function and the sigmoid function is adopted to make sure that the output ranges from 0 to 1. A causal mechanism is used to achieve real-time processing, where only the past frames are involved in the convolution calculation. The tensor output size of each layer is given with (*Channels*, *TimeStep*, *Feat*) format, which is shown in Figure 1. A more detailed description of the network can refer to Table 1. The total number of trainable parameters of the network is 0.59 M.

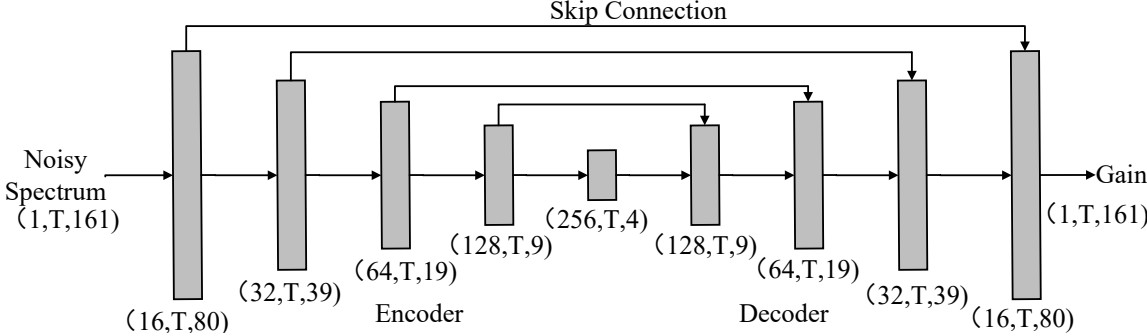

**Figure 1.** The network architecture adopted in this study. Input is the noisy magnitude spectra and output is the estimated gain functions. T refers to the timestep length of the utterances within a minibatch.

**Table 1.** Detailed description of the netowrk used in the manuscript. The input size and output size of 3-D representation are given in $(Channels, TimeStep, Feat)$ format. The hyperparameters are specified with $(Kernel, Stride, Channels)$ format.

| Layer Name | Input Size | Hyperparameters | Output Size |
|---|---|---|---|
| reshape_size_1 | $T \times 161$ | - | $1 \times T \times 161$ |
| conv2d_1 | $1 \times T \times 161$ | $2 \times 3, (1, 2), 16$ | $16 \times T \times 80$ |
| conv2d_2 | $16 \times T \times 80$ | $2 \times 3, (1, 2), 32$ | $32 \times T \times 39$ |
| conv2d_3 | $32 \times T \times 39$ | $2 \times 3, (1, 2), 64$ | $64 \times T \times 19$ |
| conv2d_4 | $64 \times T \times 19$ | $2 \times 3, (1, 2), 128$ | $128 \times T \times 9$ |
| conv2d_5 | $128 \times T \times 9$ | $2 \times 3, (1, 2), 256$ | $256 \times T \times 4$ |
| deconv2d_1 | $256 \times T \times 4$ | $2 \times 3, (1, 2), 128$ | $128 \times T \times 9$ |
| skip_1 | $128 \times T \times 9$ | - | $256 \times T \times 9$ |
| deconv2d_2 | $256 \times T \times 9$ | $2 \times 3, (1, 2), 64$ | $64 \times T \times 19$ |
| skip_2 | $64 \times T \times 19$ | - | $128 \times T \times 19$ |
| deconv2d_3 | $128 \times T \times 19$ | $2 \times 3, (1, 2), 32$ | $32 \times T \times 39$ |
| skip_3 | $32 \times T \times 39$ | - | $64 \times T \times 39$ |
| deconv2d_4 | $64 \times T \times 39$ | $2 \times 3, (1, 2), 16$ | $16 \times T \times 80$ |
| skip_4 | $16 \times T \times 80$ | - | $32 \times T \times 80$ |
| deconv2d_5 | $32 \times T \times 80$ | $2 \times 3, (1, 2), 1$ | $1 \times T \times 161$ |
| reshape_size_2 | $1 \times T \times 161$ | - | $T \times 161$ |

*4.4. Loss Functions and Training Models*

This paper chooses three loss functions including MSE in (4), Time-MSE-based loss (TMSE) [35] and recently proposed SI-SDR-based loss [35] as baselines. As a T-F domain-based network is used, an additional iSTFT layer is needed to transform the estimated T-F spectrum back into time domain for TMSE- and SI-SDR-based loss [38]. The iSTFT layer is a type of specific deconvolutional layer, whose basis function corresponds to the iSTFT coefficient matrix. The baselines are compare with the proposed generalized loss function given in (18) with (20) and (21).

## 5. Results and Analysis

This paper uses four objective measurements to analyze the performance of proposed generalized loss, including noise attenuation (NA) [25], speech attenuation (SA) [25], PESQ [17], and SDR [39].

*5.1. The Impact of $\gamma$, $\beta_0$ and $\mu$*

The testing results w.r.t. $\gamma$, $\beta_0$ and $\mu$ are shown in Figure 2, where $\gamma = 1, 2, 3$, $\beta_0 = -10$ dB, $-20$ dB, $-30$ dB and $\mu = 0.5, 1, 2, 3, 4$ are considered. Here $\alpha_0$ is set to 1 for all the conditions. The test results of three baselines are also presented as the comparison. From this figure, one can observe the following phenomena. First, the increase of $\beta_0$ will decrease NA. This is because the residual noise control mechanism is introduced for optimization, which means, during the training process, the residual noise in the estimated spectra will gradually get close to the preset residual noise threshold. As a consequence, the characteristic of the residual noise is expected to be effectively preserved, which will be further confirmed by subjective listening tests in the following. Second, the increase of $\mu$ is beneficial to noise suppression and meanwhile introducing more speech distortion. As generalized loss can be viewed as the joint optimization of both speech distortion and noise reduction, a larger $\mu$ shows that more emphasis is given on noise suppression and it leads to smaller gain values, as (17) states, where on the one hand more interference is suppressed and on the other hand, more speech components are inevitably abandoned. Third, the increase of $\gamma$ has a negative influence on NA and SA. In addition, when $\gamma$ is set to 2, it shows a better objective speech quality than $\gamma = 1, 3$.

According to the above results, we have some general guidelines to choose the three parameters including $\gamma$, $\beta_0$ and $\mu$. First, NA is expected to be as large as possible while SA needs to be as small as possible, indicating more noise components can be suppressed with less speech distortion. Second, both PESQ and SDR need to be as large as possible, indicating better speech quality is achieved.

Considering the effects of different parameters illustrated in the last paragraph, among various parameter configurations with $(\gamma, \beta_0, \mu)$ format, (2, −30 dB, 0.5), (2, −30 dB, 1) and (2, −20 dB, 1) can be chosen for practical applications. This is because relatively better performance can be achieved for four objective metrics. One can observe that the three competing loss functions can get better performance in some objective metrics, while they may suffer much worse performance in others. For example, SI-SDR has the largest value of SDR, while its PESQ score is even lower than the MSE, which is consistent with the study in [19].

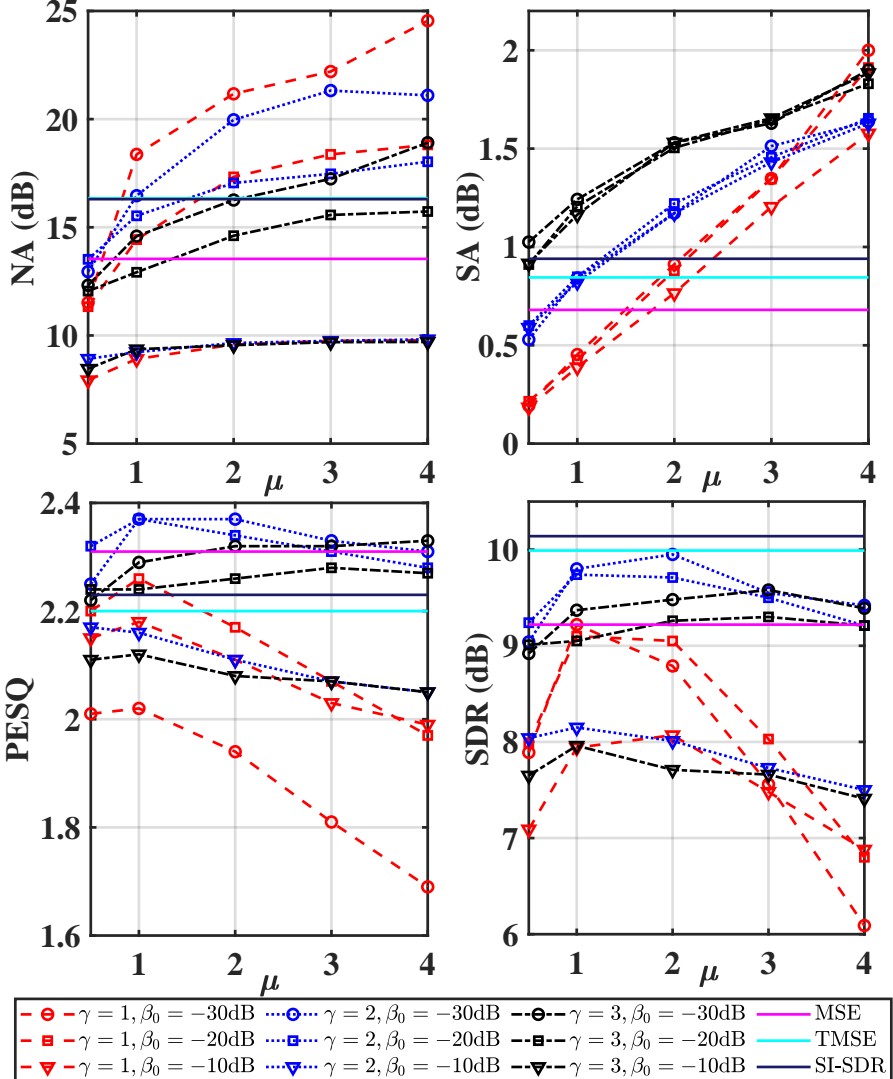

**Figure 2.** Test results in terms of NA, SA, PESQ and SDR. The averaged PESQ score of the noisy signals is 1.80 and its averaged SDR is 2.51 dB. Here $\alpha_0$ is fixed at 1 for all conditions.

### 5.2. The Impact of $\alpha$

To analyze the impact of $\alpha$, we select one type of parameter configuration, with $\gamma = 2$, $\beta_0 = -20$ dB, and $\mu = 1$, which has shown the best performance among various configurations. The value of $\alpha$ ranges from 1 to 2 with the interval 0.1. The reason for choosing $\alpha \geq 1$ is to avoid the gradient value infinite problem during the back-propagation process for $\alpha < 1$. The metric results w.r.t. $\alpha$ are given in Figure 3. One can observe the following phenomena. First, the increase of $\alpha$ will decrease both NA and SA values. This is because the estimated gain in each T-F bin ranges from 0 to 1, and the increase of $\alpha$ will lead to a larger value, cf. (17). As a consequence, the network tends to attenuate less background interference and preserve more speech components. Second, both PESQ and SDR tend to decline with

the increase of $\alpha$, which can be explained as more residual noise components are preserved, and they will heavily degrade the speech quality although the speech distortion is reduced.

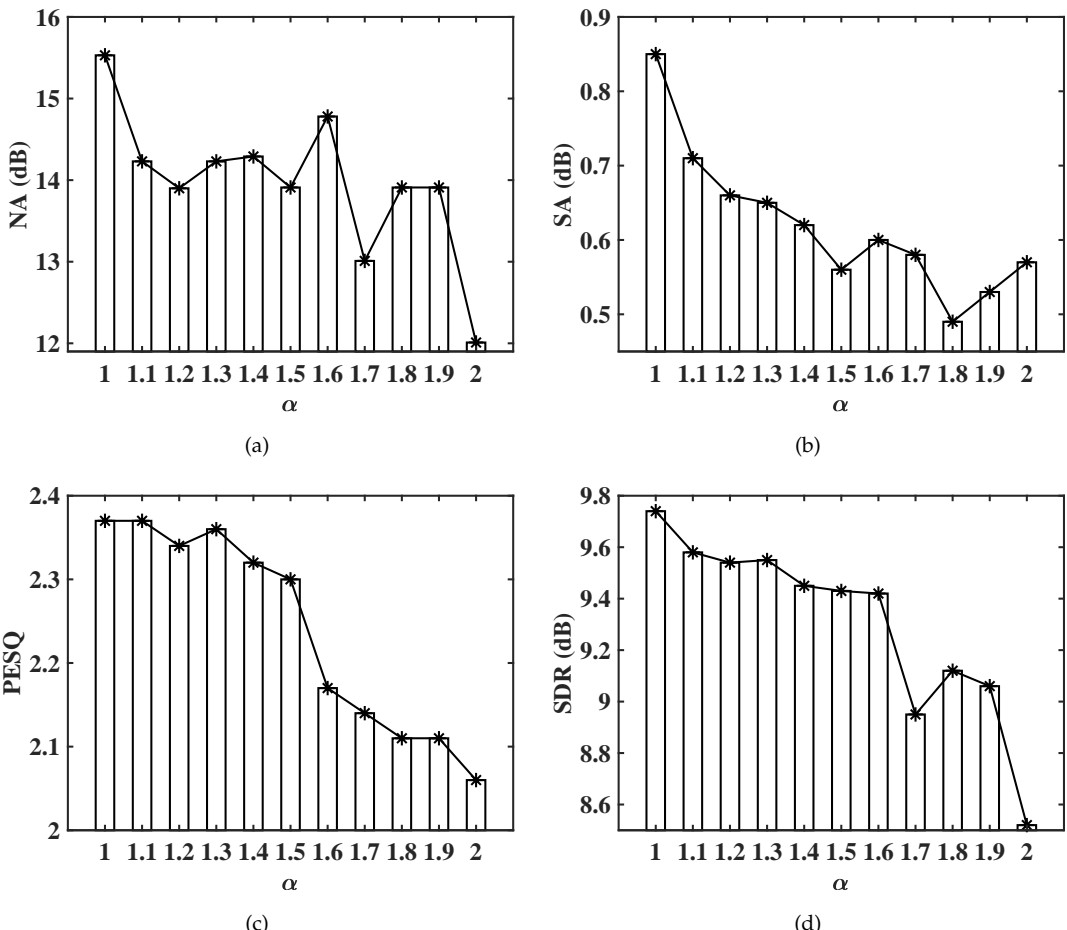

**Figure 3.** Metric scores with the increase of $\alpha_0$. (**a**) NA scores with the increase of $\alpha_0$. (**b**) SA scores with the increase of $\alpha_0$. (**c**) PESQ scores with the increase of $\alpha_0$. (**d**) SDR scores with the increase of $\alpha_0$.

### 5.3. Subjective Evaluation

To evaluate speech quality of the proposed generalized loss (GL) function, a subjective evaluation test is conducted among GL and baselines, where we follow the subjective testing procedures of [40]. In this comparison, we choose the parameter configuration (2, −20 dB, 1) and $\alpha_0 = 1$ for the proposed GL function. The experiment is conducted in a standard listening room with the size (5 m × 4 m × 3 m), where 10 listeners participate. The listening material consists of 20 utterances, each of which includes one male and female utterance selected from TIMIT corpus and is mixed with one of five noises including aircraft, babble, bus, cafeteria, and car. Four SNR conditions are selected for mixing, i.e., −5 dB, 0 dB, 5 dB, 10 dB. A speech pause of 3s duration is specifically inserted before each utterance. Then, the duration of each listening utterance is about 13s. Each listener needs to write down the utterance index that they prefer considering both noise naturalness and speech quality. The same as [40], "Equal" option is also provided if no subjective preference can be given. To overcome inertia, the utterance index in each pair is shuffled. The averaged subjective results are presented in Table 2. From this table, one can observe that the proposed GL function with residual noise control achieves better performance in subjective testing, which can be explained as the proposed GL method can effectively recover speech components while preserving the characteristic of background noise to some extent compared with all the baselines.

**Table 2.** Results of subjective listening test. The numbers indicate the percentage of votes in favor of one approach. The choice "Equal" means no subjective difference.

| Methods | GL | MSE | Equal |
|---|---|---|---|
| **Preference** | 70.0% | 22.0% | 8.0% |
| **Methods** | **GL** | **TMSE** | **Equal** |
| **Preference** | 66.5% | 22.0% | 12.5% |
| **Methods** | **GL** | **SI-SDR** | **Equal** |
| **Preference** | 70.5% | 23.5% | 6.0% |

## 6. Conclusions

This paper derives a generalized loss function which can easily make a balance between noise attenuation and speech distortion with multiple manual parameters. In addition, MSE and other typical loss functions are revealed to be special cases. Both objective and subjective tests are conducted to show that it is important to control the residual noise for supervised speech enhancement approaches, where the residual noise becomes much more natural than before. Moreover, compared with other competitive loss functions, the proposed loss function obtains comparable performance in objective metrics and much better subjective evaluation results when suitable parameter configurations are selected. Further work could concentrate on studying a combination of the residual noise control scheme with objective metrics-based loss functions to improve the naturalness of the residual noise.

**Author Contributions:** Conceptualization and methodology, A.L. and C.Z.; software and validation, A.L. and R.P.; writing—original draft preparation: A.L.; writing—review and editing, C.Z. and X.L.; supervision: X.L. All authors have read and agreed to the published version of the manuscript.

**Funding:** This research was funded by the National Science Foundation of China under Grant No. 61571435, No. 61801468, and No. 11974086.

**Conflicts of Interest:** The authors declare no conflict of interest.

## Acronyms and Abbreviations

| | |
|---|---|
| SNR | signal-to-noise ratio |
| MMSE | minimum mean-squared error |
| MSE | mean-squared error |
| DNN | deep neural network |
| PESQ | perceptual evaluation speech quality |
| STOI | short-time objective intelligibility |
| CL | component loss |
| GL | generalized loss |
| SI-SDR | scale-invariant speech distortion ratio |
| T-F | time-frequency |
| DFT | discrete Fourier transform |
| SGD | stochastic gradient descent |
| STFT | short-time Fourier transform |
| BN | batch normalization |
| ELU | exponential linear unit |
| TMSE | time mean-squared error |
| iSTFT | inverse short-time Fourier transform |
| NA | noise attenuation |
| SA | speech attenuation |
| SDR | speech distortion ratio |

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
