# Peer review of "A Supervised Speech Enhancement Approach with Residual Noise Control for Voice Communication"

_applsci, doi:10.3390/app10082894_

Round 1

Reviewer 1 Report

The paper investigates a supervised speech enhancement approach based on a suitable generalized loss function, that specifically takes into account speech distortion and residual noise, allowing to control the resulting perceptual quality. The problem is clearly introduced and the proposed formulation is derived and confronted with state-of-art literature. In the experimental section results on TIMIT are discussed: simulations with different noises at various SNRs are generated and used to train and validate a model based on the U-net architecture. The results are convincing, although the analysis could have considered also subjective evaluations on real recordings to validate the approach in realistic conditions, where other effects (e.g. reverberation) and non-linear speech-noise combination occur. Moreover, it would be useful to discuss the selection of the suitable parametrization for the problem at hand, considering the multiple parameters introduced in the problem formulation and the different metrics and trends obtained according to a specific setup.

Overall, the work is interesting, presenting a synthetic but comprehensive framework for supervised speech enhancement based on a generalized loss function. However, a (small) extension of the subjective listening test on data coming from recent datasets (e.g. data released for the CHiME-6 or the Deep Noise Suppression Challenge) would reinforce the conclusions of the proposed approach.

Minor typos to be corrected:

  • Figure 2: "the noisy signal IS 1.80",
  • line 198: "peper";

Author Response

Dear Reviewer 1

Thank you very much for reviewing and for your comments. We considered your comments thoroughly and changed the manuscript to account for all the comments. Please see the attached file Reply to "R1.pdf" for details.

Reviewer 2 Report

This paper suggests a new loss function in the supervised speech enhancements approach. The novelty of the proposed loss function is that it introduces a residual noise control to provide a good trade-off between noise reduction and speech distortion. 

Here are a few remarks regarding the submitted manuscript.

1) The introduction needs to give a more useful overview of different state-of-the-art approaches for speech enchantments.

Referencing other research is superficial; it is advisable to explain the disadvantages of other approaches better. For example, what information is lost when MSE optimization is used as opposed to a perceptually motivated approach.

2) Section 2 - Problem formulation is unclear since it doesn't address the main problem. This section could be used to emphasize the theoretical implications of the proposed approach.  Also, some explanations are redundant like h(a,b,c) is a function of three variables a,b,c.

3) Proposed deep learning U-Net need supplementary information such as the shape of the convolutional filter, number of total parameters, and some training details. Also, it is not clear what is input to network or what is iSTFT layer.

4) The concluding section is missing a clear judgment about the achieved results and their analysis/comparison with the basic method form [33]. Also, regarding practical contribution, it could be interesting to analyze the computational time i.e. time required to obtain speech enhancement for one TIMIT utterance.

Author Response

Dear Reviewer 2

Thank you very much for reviewing and for your comments. We considered your comments thoroughly and changed the manuscript to account for all the comments. Please see the attached file "Reply to R2.pdf" for details.
